# A Semantic Offsite Construction Digital Twin- Offsite Manufacturing Production Workflow (OPW) Ontology

Kudirat Ayinla[1], Edlira Vakaj[2], Franco Cheung[2], and Abdel-Rahman H. Tawil[2]

[1] London South Bank University, Department of Built Environment and Architecture, London, UK. {ayinlak@lsbu.ac.uk}

[2] Birmingham City University, Faculty of Computing, Engineering and Build Environment, Birmingham, UK
{edlira.vakaj, francho.cheung, abdel-Rahman.Tawil}@bcu.ac.uk

**Abstract.** Offsite Manufacturing (OSM) is a modern and innovative method of construction with the potential to adopt advanced factory production system through a more structured workflow, standardised products, and the use of robotics for automation. However, there have been challenges in quantifying improvements from the conventional method, which leads to the low uptake. The concept of a digital twin (DT) is useful for OSM, which enables production to be represented virtually and visually including all activities associated with it, resources, and workflow involved. Thus, essential information in the product development process such as cost, time, waste, and environmental impacts can be assessed. However, the data required to have accurate results and better-informed decision-making come from heterogeneous data formats (i.e. spreadsheets and BIM models) and across different domains. The inclusion of semantic web technologies such as Linked Data (LD) and Web Ontology Language (OWL) models has proven to better address these challenges especially in terms of interoperability and unambiguous knowledge systematisation. Through an extensive systematic literature review followed up by a case study, an ontology knowledge structure representing the production workflow for OSM is developed. A real-life use case of a semi-automated production line of wall panel production is used to test and demonstrate the benefits of the semantic digital twin in obtaining cost and time data of the manufacturing for assessment. Results demonstrated the potential capability and power of capturing knowledge for an ontology to assess production workflow in terms of cost, time, carbon footprint thereby enabling more informed decision making for continuous improvements.

**Keywords:** Offsite Manufacturing. Production Workflow · Digital Twins · Ontologies. Process Modelling.

## 1 Introduction

OSM is an aspect of design for manufacturing and assembly (DfMA) that moves most of the construction processes to a factory environment where components are assembled to form modules and then transported to a final point for assembly, usually the construction site [1–3]. The benefits of this method have been widely studied, e.g. reductions in construction time, increased quality, low health and safety risks, low environmental impact, reduced whole-life cost, and a consequent increase in predictability, productivity, whole-life performance, and profitability [2, 4, 5]. While most benefits are claimed to be the outcome of the process improvements due to the workflow in a factory environment [6], quantitative assessment of the benefits is not evident. Unlike

operations onsite that focus predominately on the organization of labour and materials, the planning of OSM is more complex involving the organisation of various production line workflows, design configurations of different workstation arrangements, different automation processes, and various levels of human intervention.

The use of DTs have the benefits of simulating processes and is capable of allowing powerful data collection to enhance efficiency in the value chain [7]. Previous development of DTs has been mainly on the use of immersive technology such as augmented reality (AR) and application of DTs with Building Information Modelling (BIM) [7]. However, these applications have been mostly focused on the design and construction, and/or operational aspects of assets with little application on the manufacturing or production stage of a building [8].

This study proposes an ontology-based digital twin for assessment of the performance of OSM. Disregarding the challenges of semantic DT application, such as the need to handle high-volume streaming of data in a semantic context, provide integration of semantic models with analytical solutions, semantically link simulations to specific use-cases, and learn semantic models over time, the use of semantic web technologies or ontologies is known for being efficient in knowledge capture and sharing, and are capable of giving intelligent real-time and context-specific data, which would be useful for design development in the OSM domain. This paper explains how the modelling of OSM workflow can be supported by automated quantitative assessment from an OSM ontology developed.

## 2 Literature Review

### 2.1 Semantic Web Technologies for Construction Digital Twins

A DT is a virtual model of the real product [9], consistent with its corresponding physical entity capable of simulating and mirroring the behavior and performance of the physical entity [8]. For OSM, a DT is a virtual digital replica of a building's physical components and production methods that collects real-world information about the physical and production line workflow via sensors and other wireless technology. The "twin" is continuously updated with data collected from multiple heterogeneous sources across different construction domains and provides valuable insights about the performance, operation, or profitability using advanced analytics, machine-learning algorithms and artificial intelligence (AI). As such, a DT can serve as the backbone for OSM and as a more significant means for improving offsite construction efficiency.

A DT for the modelling of OSM production workflow needs to consider several aspects ranging from physical components (e.g. Buildings machine tools, part types to be produced, etc.) to production methods (e.g. process plans, production logics, etc.), from production workflow (e.g. placement of production resources in the factory layout) to organizational management (e.g. roles of the involved actors), from costs (e.g. labour, nominal power consumption of a machine tool) to dynamics (e.g. evolution of the states

of a resource) with data generated and captured across the entire product lifecycle. Thus, the effectiveness of a DT relies on the robust construction of intelligent services and framework to be put in place (e.g., simulation, prediction, forecasting) to support the various heterogeneous systems and technologies involved in construction [10].

Emerging Building Information Modelling (BIM) tools and technologies have changed the way information about the built environment is created, stored, and exchanged between involved stakeholders [11]. When completed, the computer-generated BIM model contains precise geometry and relevant data needed to support the construction, fabrication, and procurement activities needed to realize the building [12], and even more data on the time schedule, cost estimation, and maintenance management [13]. The use of BIM models has not benefited from real-time data inputs of the object data from OSM, as the focus in practice has been on improving the design collaboration, construction activities such as logistic management as well as operational and management of an asset. However, BIM lacks semantic completeness in areas outside the scope of the components and systems of a building. Thus, the need for an all-inclusive, sustainable approach that considers dynamic data at different levels [7]. In order to enable and encourage this exchange, a common schema has been developed, which is specifically referred to as Industry Foundation Classes (IFC). Since the advent of the IFC, more integrated methods to share construction data have emerged and have since become adopted industry-wide. At the same time, digital technologies across the board are advancing at an ever-increasing pace, taking advantage of the Internet of Things (IoT) and Artificial Intelligence (AI) agents (data analytics, machine learning, deep learning, etc.). The success of a DT relies on the various processes and data layers that are intended to support construction intelligent services and applications assuming a robust framework is in place to support the various heterogeneous systems and technologies involved.

The evolution of BIM should be carefully framed within a paradigm that factors people, processes, and other emerging technologies in an increasingly inter-connected world through the application of sensor networks [13]. Building/infrastructure-related information can be directly or indirectly integrated within available digital technologies in a BIM-enabled environment, a broad list of related research work is detailed in [14].
The use of semantic models (ontologies) as demonstrated by IFC for openBIM is particularly useful as it links data of many contexts. The DT paradigm aims to enhance existing construction processes and BIM models, with their underpinning semantics (e.g., IFC, COBie) within the context of a cyber-physical synchronicity, where the digital models are a reflection of the construction physical assets at any given moment in time [13]. The current limitation is that the data shared from IFC is only based on the geometry of a building while COBie data is operational. There has been little focus on modelling the knowledge of the manufacturing and production aspects with regards to the use of OSM. This sort of data is not captured in a BIM model thus limiting the potential interventions in terms of optimizing processes and increasing efficiency during the manufacturing stage of building assets. Given that a DT continuously receives

data from different sources, there is a need to develop proper ontologies for data representation and formalization.

## 2.2 Semantic Digital Twin Representation through Ontologies

Ontologies are a well-established approach for leveraging data and information sources with semantics, thus providing a shared, machine-understandable vocabulary for information exchange among dispersed agents (e.g. humans and different machines) interacting and communicating in a heterogeneous distributed intelligent system [15]. There have been previous studies on the application of domain ontologies for supporting data capturing in DT. Chevallier et al. [16] propose to build a Smart Building Digital Twins reference architecture that is based on various domain ontologies such as ifcOWL for the infrastructure, SSN (Semantic Sensor Network), and SOSA (Sensor, Observation, Sample, and Actuator) for IoT description and Vakaj et al. [17] developed the Offsite Housing Ontology to support offsite housing design evaluation. DTs are independent of tools and servers where each IoT is associated (linked) to its physical counter object. Using ontologies, all the information produced by sensors, which reflect the state of a Smart Building over time could be associated with their physical ifcOWL counterparts.

The review of existing literature in the OSM domain was conducted to reuse terminologies and existing knowledge classification. The review also considered the possibility of extending some of the existing ontologies relevant to the research problem. Ontologies such as ifcOWL ontology generated from the IFC standards [18], Building Topology Ontology (BOT) [19] describing the topology of buildings, and Building Product Ontology (BPO) [20] for describing building products, are very useful for modelling the AEC domain information in a Linked Building Data format. However, BOT and BPO ontologies were purposely implemented as lightweight ontologies to promote re-use and do not include specific DfMA concepts for offsite manufacturing which is a challenge. Similarly, while MASON [21] provides the core concepts of manufacturing, extending it to include the complexities and depth of analysis of buildings, and, more so, offsite buildings, creates a substantial challenge with redundancy and complexity. It is necessary that any extension of an ontology leads to a result that is lightweight, efficient, and conceptually coherent, in order to support adoption and implementation. As argued by Kalemi et al. [22], ontologies in complex domains that attempt to be all-inclusive often are not optimal for purpose: a prominent example in construction is the development of BOT as a way of addressing ifcOWL's complexity.

Hence, the study further complements these ontologies by modelling low-level concepts relating to the production stage of an OSM building workflow. The semantic DT approach proposed in this study provides a viable way of crossing from a BIM worldview with its existing ifcOWL knowledge domain, towards a holistic view which promises greater possibilities by the intersection of production knowledge through descriptive and formal domain models using ontological inferences in real-world situations. Given that a DT continuously receives data from different sources, there is a need to develop proper ontologies for data representation and formalization, and it will be

beneficial for the DT to also incorporate data on production workflow for monitoring factory shop floor efficiency as illustrated in Figure 1.

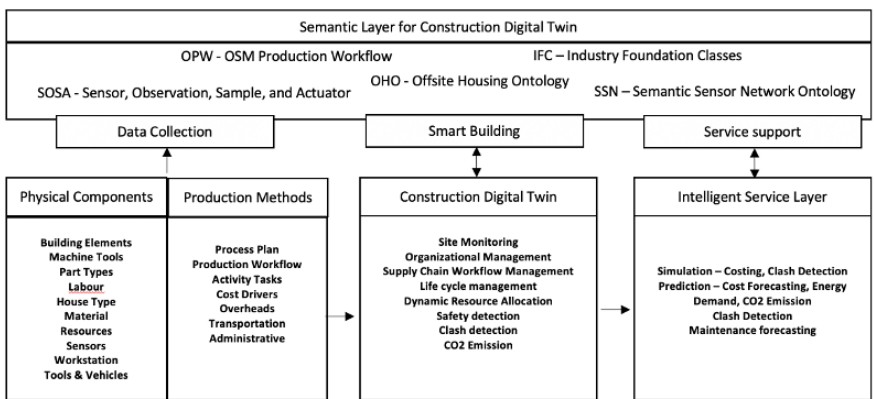

Fig. 1. Digital Twin Semantics for Smart OSM Construction Services

## 3 OSM Production Workflow (OPW) Ontology

The OSM production workflow (OPW) ontology aims to model the knowledge of the production process of a factory-manufactured building from the point of material delivery to the transportation of the finished manufactured products to the site. The data gathering process is based on the case study of OSM house production involving various units of analysis (i.e. the cases of two production methods, static production and semi-automated linear production methods of factory house building). The multidisciplinary knowledge required to define the main concepts and their relationships is collected from different sources as illustrated in Figure 2.

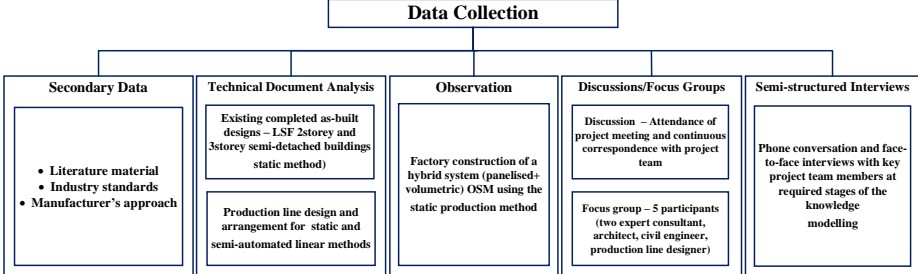

Fig. 2. Knowledge collection strategy

The data collection is followed by the ontology development exercise. The OPW ontology design and development is guided by the Meth-ontology methodology, which is considered one of the most mature methods for ontology development [23, 24]. Four stages were followed comprising:

– Specification: identifying the purpose of ontology and defining some competency questions.
– Conceptualization: capturing domain knowledge via various sources of information such as literature review and case studies, and informal representation of knowledge gathered.
– Formalisation: knowledge representation in semi-formal languages ready for implementation.
– Implementation: formal representation of the knowledge in an ontology using a machine-readable language (OWL).

A set of competency questions are defined and used to guide the knowledge modeled in the ontology. Relevant knowledge and data to answer those queries in the ontology are semantically modeled using the OWL. For the proposed OPW ontology, the set of competency questions include:

– What activities are involved in manufacturing a house using various systems of OSM (i.e. panelised, volumetric or hybrid methods) and what resources are involved in each process? (See Experiment 1);
– What is the hierarchy of events and process flow based on the factory layout, and which activities fall in each workstation and production methods? (See Experiment 1);
– What are the time and cost spent on each activity and ultimately workstations involved in producing a house using the OSM method? (See Experiment 2);
– What is the proportion of waste generated from activities involved in the production process of different methods? (See Experiment 3).

## 3.1 Classification and Relations in the OPW Ontology

For the OPW ontology, there are 8 major classes (Level 1) required to formalise the production process knowledge which relates to all offsite methods (Figure 3). These include concepts such as (i) OSMFactoryProductionMethod – for classifying all types of production systems (ii) Production Process – for classifying the processes involved in each method, (iii) WorkStation – for capturing the categories of activities in each station (iv) ProcessType – relating to the workflow of events in the production process, (v) Activity – for classifying the major tasks performed on the factory shop floor, (vi) Resources – relating to resources consumed in the processes (vii) Product – relating to the final product from the production line and (viii) Building – for classifying the final product at the destination point, which is onsite.

The subclasses of the major classes are represented (Level 2) with the *isA* relationship to denote a parent-child relationship.
Finally, some relationships between the various classes are represented. The key attributes/properties needed to include semantics in the ontology include data properties such as Cost, Time, Distance, Length, Width, etc., and object properties such as *hasComponentPart*, *consumes*, *isComposedOf*, *hasOutput* etc.

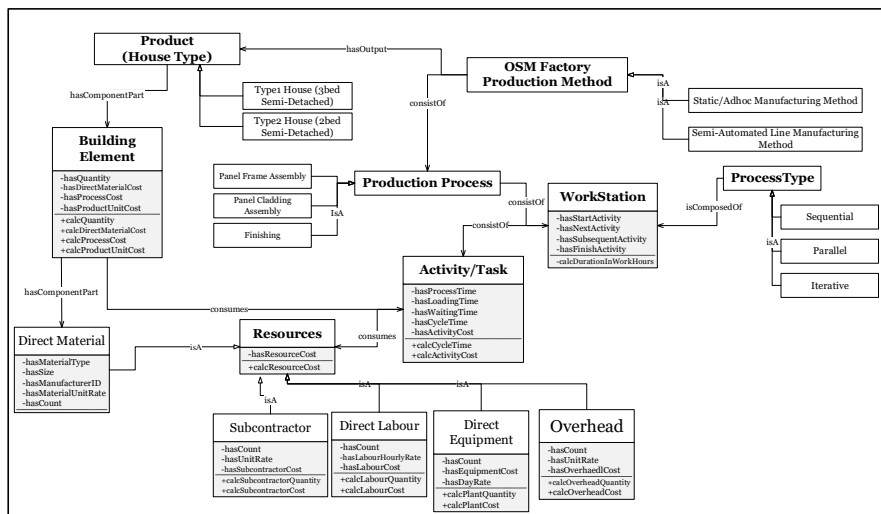

Fig. 3. Classes, subclasses, and properties of the OPW ontology

As an example, an OSM production method (OSMProductionMethod) '*consistOf*' production processes (ProductionProcess), and each production process (ProductionProcess) '*consistOf*' various work station (WorkStations). There are different sequences of events in each workstation (ProcessTypes) which could be parallel, sequential, or iterative in nature. Also, each work station (WorkStation) '*consistOf*' activities (Activities), and activities (Activity) '*consume*' resources (Resources) which can be labour, plant/equipment, materials, or overhead. Finally, the production methods (OSMProductionMethod) have the products as outputs (Product). These classes, attributes, and relationships will enable retrieval of data on the instances in the ontology to support analysis of the production workflow for design making and continuous improvement. The OPW ontology developed is published on the web for sharing, and reuse of the production knowledge relating to OSM.[1]

## 4    Use Case of a Static and Semi-automated Linear Methods of OSM Production

Having finalised the knowledge modelling in the ontology, a use case of a type of OSM production process was selected to enable the population of the ontology with instances and retrieval of data, i.e. a semi-automated linear method of factory house building. The data used for the workflow and activity modelling are based on an actual project using the static method and the design and simulation of a semi-automated production process by a partner production engineering company.

---

[1] The OPW ontology can be accessed from: https://edlirak.github.io/oho-pro/index-en.html

The workflow of the static and semi-automated linear method used in the use case is illustrated in Figures 4 and 5 respectively. The static method involves an ad-hoc manual production sequence predominantly dependent on labor resources while the semi-automated method features a structured workflow where production is done on an assembly line with dedicated stages/stations. The production of a wall panel is composed of three stages, the first being frame assembly and another for cladding assembly. The third stage involving finishing is to be done manually for both methods. The semi-automated consists of automated machines such as various robotic arms, and some human interventions and tasks embedded in the workflow. The third stage involving applying finishes is to be done manually. The completed units are moved on a conveyor system and are picked up by fork-lift or trolleys to be loaded on transport vehicles. The batch manufacturing method is used instead of a singular house build method where the tool is set up for a particular batch type of frame at a time. Finished batches of wall panels are moved to a temporary storage area in the factory and later transported to site. The use-case selected for testing the ontology is a case of a 3BED Semi-detached house type (hereafter House-Type 1). House-Type 1 is made of Light Steel Framed (LSF) material using the panelised system of OSM. For the factory production, the external frame of the house is divided into a total of 32 panels which are the output from the production process. This consists of 20 external clad panels and 12 internal panels for the party walls.

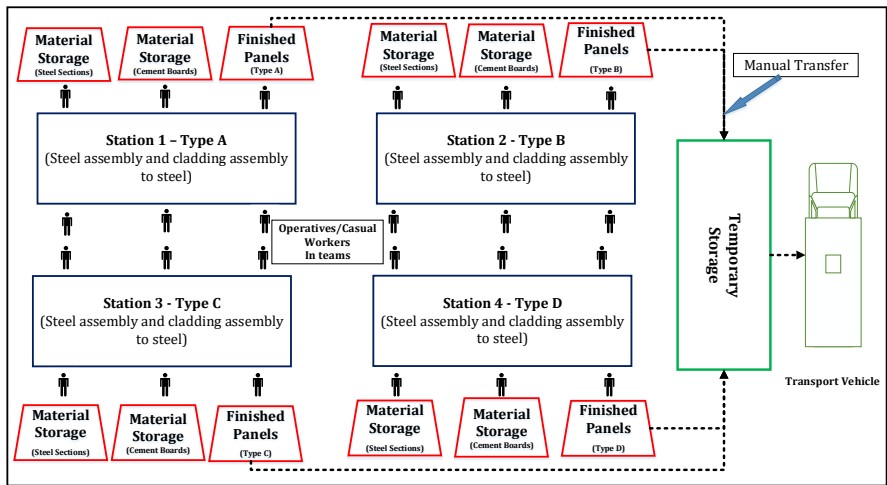

Fig. 4. Static production arrangement

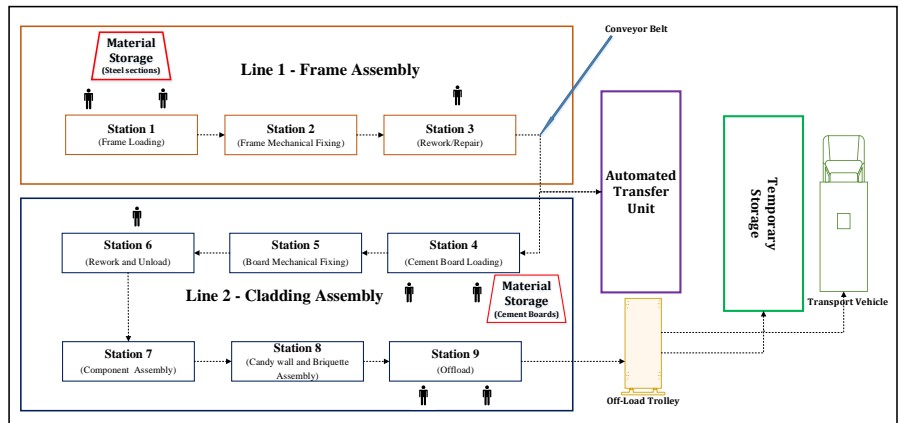

Fig. 5. Semi-automated linear production arrangement

## Experiment 1: Querying the Ontology for Retrieving General Information Relating to the Production Line

The first experiment demonstrates the various types of data that can be retrieved from the semantic model regarding the production process of a building element. In this test, the semantic model is queried to generate data on the activities involved in the production of an instance of a wall panel (i.e. *3BED-GF-Front-LSF-01*) and the resources consumed in the process (Figure 6). The query returns data about the production process that can enable understanding of the processes and the consumption of resources. This information can potentially compare various methods of production for the same building type in terms of workflow, chain of events, and performance.

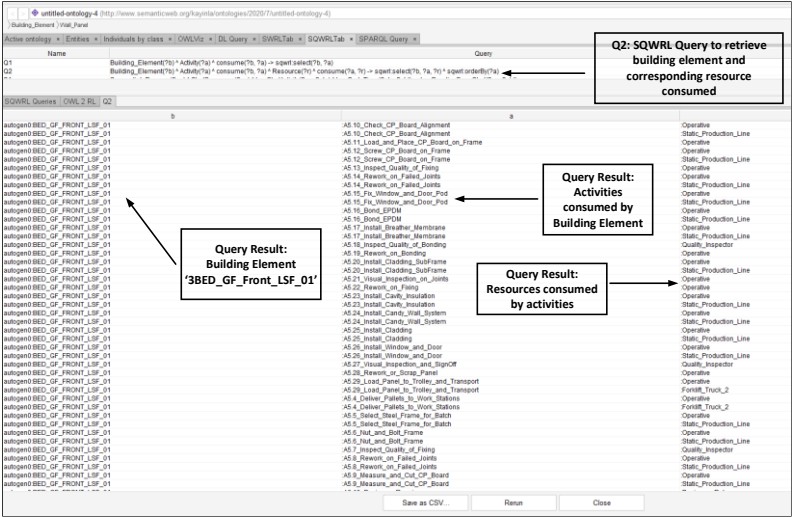

Fig. 6. SQWRL Query Result - Process Information for an instance of wall panel

## Experiment 2: Retrieving and Analyzing Cost Information of Products

The second competency question relates to retrieving information on the cost of activities involved in an OSM production process and linking these with the various building elements that consume the activities.

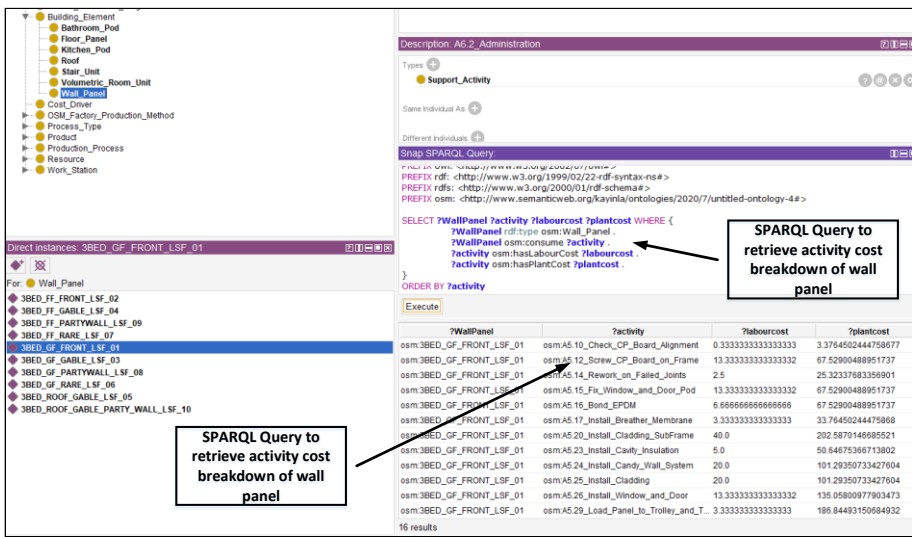

Fig. 7. SPARQL Query Result - Activities cost of an instance of wall panel with the static production method

The building elements are in turn related to a specific house type through the object property '*hasComponentPart*' thus allowing for the cost of each product to be computed. The data property relating to this is the '*hasActivityCost*' which is computed by summing up the cost of resources consumed by activity through the properties '*hasLabourCost*', '*hasPlantCost*', and '*hasOverheadCost*' depending on the resources applicable to each activity. The activity costs in turn form the process cost in producing any product from the OSM method. The data properties ('*hasLabourCost*', '*hasPlantCost*', and '*hasOverheadCost*') are computed with the help of SWRL rules and are then fed back into the knowledge base as inferred properties. To test the ontology, a query was developed to retrieve information on the breakdown of the cost of activities involved in the production of a type of wall panel with both methods of production, using the instance of '*3BED-GF-Front-LSF-01*'. The query result returned data on the cost of each activity based on the labour and plant consumed in the production of the wall panel instance (Figure 7). This information can be useful for the manufacturer in analyzing the process cost of any building element while reviewing which activities consumes the most resources and why based on two alternative approaches.

## Experiment 3: Analysing Cost and Time Spent on Processes in Various Production Methods

The third selected test case allows the analysis of the time spent on the various categories of activities between two methods of OSM production, the static and semi-automated methods, and analyzing value-adding in terms of time and cost. The aim is to compare the process information for each production method.

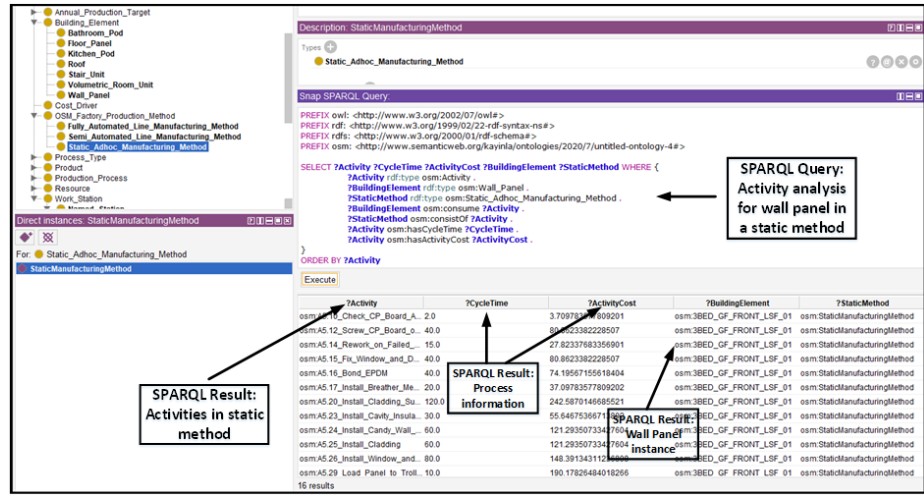

Fig. 8. SPARQL Query Result - Activities cost of an instance of wall panel with the static production method

As the ontology already contains knowledge on the two methods and the sort of activities involved, this will allow the manufacturer to analyse both options in aspects such as the time spent on various activities in a product development process and the cost incurred. Potentially also, to determine where intervention is needed for continuous improvement.

A SPARQL query was written to retrieve information on the cost and time spent in the production of the wall panel instance *'3BED-GF-Front-LSF-01'* for both methods of OSM production. Figure 8 shows the result for the static method while Figure 9 shows the results for the semi-automated linear method of OSM production.

The query result returned data on the cost and time of each activity consumed in the production of the wall panel. This information can be useful for the manufacturer in analyzing the efficiency of the various methods of production and in determining the output/productivity that can be attained.

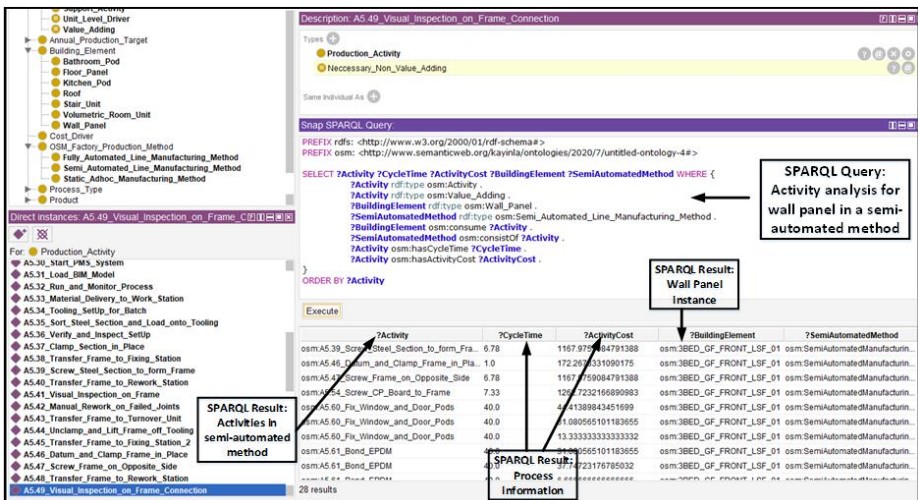

Fig. 9. SPARQL Query Result – Activities cost of an instance of wall panel with the semi-automated production method

# 5    Conclusions

This paper shows a newly developed OSM Production Workflow (OPW) Ontology and how it is applied to obtain knowledge from the ontology to evaluate processes. It demonstrates how semantic technologies can be applied to link production data to offsite building components. OPW can complement widely adopted data exchange schema such as IFC as the latter focuses on geometric data exchange by adding another dimension of knowledge relating to production workflow.

The linkage between production data and building elements is a novel development of semantic DT, addressing the manufacturing aspect of the building life cycle that has not been widely explored. The implication is significant as the use of ontology enables multiple usages of a single data source. OSM production can be queried, monitored, and improved continuously over time. This will allow the development of a variety of applications to be used in relation to production, e.g. measuring efficiency or optimising modular product and processes, and so on.

# Acknowledgments

This research was supported by UKRI COVID-19 Extension Grant "DfMA house Part 2". The authors would like to thank the participating industrial partners of the research project, particularly Hadley Group, Walsall Housing Group, and QM Systems for their contribution to this study

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
