# OpenReview forum: "A Semantic Offsite Construction Digital Twin- Offsite Manufacturing Production Workflow (OPW) Ontology"
_eswc-conferences.org/ESWC/2021/Workshop/SeDiT — SeDiT 2021 Oral_

### Official Review · AnonReviewer1 · 2021-03-16
**A Semantic Offsite Construction Digital Twin- Offsite Manufacturing Production Workflow (OPW) Ontology**

**Rating:** 5
**Confidence:** 4

**Review:**

This article proposes an ontology for Off-Site Manufacturing (OSM) processes, which can later be used for supporting OSM workflows. OSM is a topic of interest in the Architecture, Engineering and construction domain, and the combination of Digital Twins and Ontologies to address its challenges is an innovative and interesting approach. Although the topic fits well in the scope of the workshop, the article needs to be revised by authors as there are many mistakes that could easily be fixed by using a spell checker, as well as sentences and whole paragraphs that are repeated. More importantly, the proposed ontology is not available online, or at least authors did not indicate where to find it. Therefore, not only cannot the ontology be downloaded, but also the quality of the ontology cannot be evaluated and or reused by users. Summarising, although I think the article is interesting, since the ontology is its core part and it is not available online, I cannot give a score higher than 5.

Some comments that should be addressed to make the article acceptable:
- The ontology should be available online.
- Authors did not even analyse existing ontologies to see if the knowledge to be modelled by the OPW already existed and they could be reused.
- More information on the documentation and metadata of the ontology should be provided.

Some other minor comments that could be used to improve the article:
- The last paragraph of the Introduction section should explain how the article is structured. For example, what the reader may find in each section.
- The use of the Methontology methodology is not sufficiently justified. There are many more recent methodologies, which also have a big adoption (e.g., NeOn), that were not considered.
- Why are both SQWRL and SPARQL used?
- Section 4 is labelled as Section 1 (page 7)
- Some words are repeated (e.g., such as such as in the Abstract)
- Some sentences are repeated (e.g., the use of semantic models (ontologies) is particularly useful... in pages 3 and 4)
- A whole paragraph is repeated (The use of semantic models (ontologies)... in page 4)
- Some sentences are incomplete (e.g., DTs and are independent from tools in page 4)
- Once an abbreviation is explained once, it does not need to be explained again (e.g., OWL)
- There are many spelling mistakes (e.g., serviceas instead of services or predication instead of prediction)

---

### Official Review · AnonReviewer2 · 2021-03-25
**Lack of review of the existing literature**

**Rating:** 5
**Confidence:** 4

**Review:**

The objective of the paper is to develop an ontology that represents the workflow of Off-site manufacturing processes for the construction industry. The paper state that the representation of those production processes can enhance the assessment of important variables such as time, cost, and waste produced, leading to better decision making. The ontology has been constructed following the Methontology methodology, and it has been tested by means of a use case to verify that the competency questions are being addressed.

In general, the main problem that I found with the paper is the lack of review of existing models that also tried to represent the domain of manufacturing. From a fast review in google scholar I could found the following models that more or less talk about the same topic:

* MASON ontology.
* PRONTO Ontology.
* ONTO-PDM ontology.
* MRO ontology.

Also the lack of alignment with other well-known ontologies for concepts related to the building domain. For example the BOT ontology, ifcOWL ontology, etc. The document also contains sentences and entire paragraphs repeated, along with grammatical typos. Finally, another important point is the lack of documentation or repository with information related to the ontology in order to check the whole implementation.

Major changes:
* Include a more detailed state-of-the-art section with ontologies related to the domain being addressed.
* Publish the ontology and the documentation.

Minor changes:
* Figure 1 caption has errata: Serviceas --> Services
* Paragraph before figure 1 on page 4 is repeated.
* In experiment 1, page 8:
   "The query returns data about the production process that can be enable understand of the processes and ...".
   Change it for:
   "The query returns data about the production process that can enable the understanding of the processes ..."

* On page 9:
   "This information can potentially to compare various methods of production ..." change it for:
   "This information can potentially be useful to compare various methods of production ..."

---

### Official Review · AnonReviewer3 · 2021-03-28
**Lack of concrete use case example**

**Rating:** 6
**Confidence:** 4

**Review:**

This paper proposes an ontology in Off-Site Manufacturing domain: OPW, using the Meth-ontology methodology. This ontology aims to model knowledge of production process, including variables such as cost and time spent, to analyze different production methods. Some example are presented to validate the use of this ontology.

This paper is interesting, with a real application domain, and tests that prove the utility and the feasibility. However, there are some things that could be improved. First of all, I would suggest the use of concrete use case example, such as in page 7 to explain classes and their relationship, and for experimentations too. I would also suggest to publish the ontology.

Some other issues:

Page 6: Part 2.1 => Should be 3.1 or just an independant part (there is no part 3.2)

Page 7: Part 1 => part 4 //
As said before, a concrete example could be added to illustrate the explanation before part 4

Page 9: Figure 6 is not readable. The SQWRL query + a zoom on results (and maybe an explanation of these results) could be more relevant

Page 11: It could be relevant to explain a result line with the concrete use case example

---

### Decision · Program_Chairs · 2021-04-08

Accept (Oral)